# Possibility of Utilization of Gate Geometry to Modify the Mechanical and Structural Properties of Castings on the Al-Si Basis

**DOI:** 10.3390/ma13163539

**Published:** 2020-08-11

**Authors:** Jan Majernik, Stefan Gaspar, Jan Kmec, Monika Karkova, Jozef Mascenik

**Affiliations:** 1Institute of Technology and Business in České Budějovice, Okružní 517/10, 37001 České Budějovice, Czech Republic; kmec@mail.vstecb.cz (J.K.); karkova@mail.vstecb.cz (M.K.); 2Department of Technical Systems Design and Monitoring, Faculty of Manufacturing Technologies with Seat in Prešov, Technical University of Košice, Bayerova 1, 08001 Prešov, Slovakia; stefan.gaspar@tuke.sk (S.G.); jozef.mascenik@tuke.sk (J.M.)

**Keywords:** HPDC, aluminum alloys, modification of structure

## Abstract

The mechanical properties of die castings correlate with the inner structure of a casting, distribution of the eutectic phases, and with the content and distribution of porous cavities in the casting volume. This submitted paper deals with the issue of the possibility to influence the basic parameters affecting the quality of castings through structural modifications of the gating system. The structural parameter under assessment is the gate height. In the case of the diverse height of the gate, five sets of castings were produced. The individual sets of castings were subjected to examination of selected mechanical properties, i.e., of permanent deformation and surface hardness. At the same time, the individual sets of castings were subjected to metallographic examination of the eutectic structure of the casting. It was proved that the gate height influences the aforementioned properties of the castings and significantly affects the ratio of the eutectic phases in the volume of the casting. The conclusion describes the mutual correlation between the gate structure, the mechanical properties of the casting, and its structural composition.

## 1. Introduction

Die casting allows production of thin-walled castings with high geometric precision, positive mechanical properties, and of low price. Increasing usage of castings made of aluminum alloys and high quality requirements are the reason for the possibility to predict the properties of castings already in the process of design of the casting and gate system structure. Prediction of mechanical properties of the casting when designing the structure of the gate system and the technology of production should be based on the relations between the mechanical properties of the alloy with the relevant parameters of crystal structure and fractions of gas bubbles which are bound to porosity. Decreased quality caused by air retention in the volume of the castings becomes evident especially when the mechanical properties and machinability are decreased [1,2,3].

It is obvious that the mechanical properties of castings depend mainly on the porosity and structure of the castings. In general, a fine structure of castings provides them with better mechanical properties. The surface hardness of castings depends especially on structure. The grain size in the casting depends on cooling speed or on the level of melt undercooling during contact with the mold face. It has been proved that the high pressure die casting (HPDC) process includes abrupt temperature changes on the mold surface in the course of casting cycle change. Such behavior leads to sharp heat drops on and below the mold surface [4]. When melted alloy enters the relatively cold mold cavity, the speed of solidification highly depends on the interfacial heat transfer within the mold as well as in the molten alloy and consequently influences the microstructure and mechanical properties of the final casting. In other words, correct cooling speed could lead to preferred formation of fine microstructure in the produced castings. Increase of cooling speed results in decrease of the primary size of the Si particles as well as in shortening of distance of the dendrite’s second shoulder. On the other hand, extreme high speed of solidification can cause premature solidification of molten metal prior to filling completion. Solidification speed significantly influences the formation of microstructural defects and the mechanical properties of the final product. Except for the aforementioned, the conditions of solidification could affect the morphology and amount of externally solidified crystals (ESC) or cold flakes within the casting [5,6]. Cooling speed thus significantly influences the structure improvement and the volume ratio of the porosity within the casting. The increase of cooling speed causes a decrease of the porosity fraction at all times independent of the chemical composition of the alloy, the efficiency of the performed refinement of solid and gaseous impurities, and the modification of the collision between the α phase and silicon in the eutectic environment [3].

Porosity itself caused by retention of air and gases in the volume of the casting can be reduced in many ways. Currently, the factor frequently discussed and affecting the reduction and compression of pores is the holding pressure. It has been proved that the increased values of pressure have positive impact on the distribution, size, and the volume of pores and on the tightness of the casting. When the gate becomes solidified, the volume of pores slightly increases which is caused by solidification shrinkage of the melt. On the other hand, extreme increase of the holding pressure shortens the service life of the mold [7,8]. From the technological point of view it is possible to reduce the retention of air and gases by means of the appropriate setting of the input parameters of the die casting cycle which affect the size and distribution of the pores in the volume of the casting and the filling mode of the mold cavity. The filling mode depends on the melt flow speed when passing through the gate. It is directly proportional to the plunger pressing speed in the filling chamber. Several scientific theses have proved a correlation between the plunger pressing speed and the porosity of castings or the retention of air during the filling phase. Higher pressing speed changes the character of the melt flowing in the sprue from the laminar non-planar one—planar one to the turbulent—which causes discontinuity of the melt flow. By means of the aforementioned a continuous and regular flow face can be achieved along the entire cross section of the sprue without retaining air and gas in its volume. On the other hand, it is supposed that prolongation of the die casting cycle at low pressing speed results in reduction of the melt temperature due to the long duration of the die casting cycle which leads to other defects such as cold laps and weld lines [2,9,10].

The structure of the gating system itself performs a significant function in reduction of gas retention in the melt volume and in decrease of porosity. The die casting process can be modified and the occurrence of defects can be reduced when both the gate and venting system are changed on the level of structural design of the mold according to the melt flow observed in the mold [11,12,13]. It is obvious that the porosity values and the amount of gas retained in the melt volume are influenced by the correlation of several structural nodes within the gating system beginning with correct gate design, sprues, overflow basins, and venting channels. Therefore elimination of air retention during the first phase of designing the geometry of sprues is desired [14].

According to the aforementioned, with regards to the mechanical properties, porosity, and fineness of the crystals it is possible, to a certain extent, to influence the quality of the castings by setting the technological parameters of the die casting cycle and by a suitable structure of the gating and venting systems. In the technological preparation of casting production, however, metallurgy and melt preparation must not be omitted. The mechanical stress of castings requires A consistent, high- quality structure with maximal toughness and minimal non-homogeneity. Therefore, the structure becomes A significant quality indicator of THE aluminum castings which is closely related to the individual structural factors such as α-phase, morphology (shape) of silicon and the number and distribution of intermetallic phases [15]. In practice the modification of alloys is one of the most frequently used and widespread methods of improvement of the mechanical properties of castings on the level of metallurgical preparation of the melt. The term modification means achievement of optimal shape of eutectic silicon. Modification causes shift of the eutectic point towards higher concentrations of Si and towards lower temperature. Modification changes the size and morphology of the Si crystals. The change of eutectic silicon shape through modification from the plate to a bar or a fiber causes an increase of strength and plastic properties as well as toughness. Improvement of the mechanical properties becomes more obvious with higher content of Si in the alloy. In the case of silumin with a Si content lower than 5%, modification loses practical significance [3,10,16,17]. Eutectic silicon crystallizes during formation of eutectics to a thick plate morphology which is mechanically disadvantageous as the sharp angles concentrate the tension causing breakage when using the casting. Content elements most commonly used are Na, Sr, or Sb because even several hundred ppm of content elements can modify the alloy morphology causing a positively refined fiber structure of the eutectic Si by means of which the structure and mechanical properties of the aluminum alloy on Si basis improve [18,19,20].

The casting quality depends on the mechanical properties which are conditioned by the casting structure. The casting structure depends on the alloy preparation and on the content of the retained air, i.e., on porosity. With regards to the aforementioned facts it can be assumed, that porosity depends on the gating system structure and on the setting of the technological parameters. The morphology of the structural phases of the casting depends on the metallurgical preparation of the alloy, especially on the modification. The submitted paper deals with the issue of the possibility to influence the mechanical properties and structural composition of the castings by structural modifications of the gating system or of its parts. At the same time, the mutual correlation of the gating system structure, of the technological parameters of die casting, and of the mechanical and structural properties of the castings are examined. The variable height of the gate was selected to be the assessed parameter. Five sets of castings were die cast in the case where the influence of the gate height on the change of the selected mechanical properties, porosity of castings, and change of eutectic structure of the casting was subjected to examination. It was proved that the gate height influences both quality and quantity properties of castings and its change can influence the structure of castings. On the basis of the performed experiments it was also proved that the structure of the casting correlates with the mechanical properties, and the ratio of eutectic phases in the casting influences its surface hardness. Since the technological parameters of the die casting cycle as well as the metallurgical preparation of the melt were maintained on a constant level and the only variable parameter was the gate height, it can be assumed that by means of the correct design of the gating system and especially of the gate it is possible to modify the mechanical and structural properties of mutually correlating castings. Through examination of the causes of the change of the eutectic structure of the casting by utilization of a simulation program MAGMASOFT 5.3, it was proved that the change of gate height also influences the technological parameters of the die casting process.

## 2. Materials and Methods

The chemical composition of the alloy was verified in the laboratory by means of a spectrometer Q4 TASMAN. The ambient temperature during the test reached the value of 22 °C and the relative air moisture content reached a level of 50%. Measuring was performed with the test samples and evaluated through the average of three sparks. The measured values of chemical composition are shown in Table 1.

Experimental examination was carried out with a series of castings of an electric motor flange (Figure 1) made of alloy EN AC 47 100. The measurements were performed in the proximity of the structural hole of the casting. The location was considered to be critical with regards to further machining and mechanical stress after the casting was fixed into the electric motor setup.

Castings were die cast in a quadruple mold. The gating system, the structural design of the connection of the casting to the gating system by means of a gate and its shape are shown in Figure 2a,b.

The structure of the gate stems from the methodology described in publication [21]. The gate length for the respective type of casting was constant following the aforementioned publication and also its methodology of structure of gate connection to casting with cylindrical area. From the point of view of structure the only parameter to influence the filling mode of the mold shaping cavity is the gate area **S_G_**. Since the gate area is in this case a function of width and height of the gate and the width is constant according to [21], the gate height b_n_ was selected to be the examined parameter (Figure 2c).

Five sets of castings were die cast with variable gate height. Table 2 presents the assessed variables and the constant structural parameters of the gates.

Due to the assurance of the relevant results pointing out the independent influence of the gate on the examined properties, the individual series of the castings were produced with constant setting of the technological parameters of the die casting cycle. Table 3 shows the values of the setting of technological parameters.

Surface hardness, permanent deformation, and porosity of castings were selected to be representative qualitative properties of casting. Metallographic examination of the structures of the castings was focused on evaluation of the eutectic structure with regards to the percentage share of α and β phases in the eutectic.

A static pressure test was carried out with the equipment TIRAtest 28200 (Košice, Slovakia). Measuring was performed in accordance with GME 06 007 and GME 60 156 standards. The initial loading force was set to a value of F_a_ = 16 kN, force after the release reached the value of F_m_ = 8 kN and the loading speed reached the value of 10 mm.s^−1^ (The measurement was performed in Košice, Slovakia).

Measurement of hardness of samples was carried out with the Brinell method using the hardness tester HPO 250. The measurement conditions were specified in accordance with the STN EN ISO 6506-1 standard. The input values were as follows: intendor diameter D = 2.5 mm, loading force F = 613 N, loading time t = 10 s.

Assessment of porosity f was realized with the samples approximating with their values of permanent deformation to the arithmetic average of deformation intended for the respective set of samples. Analysis of porosity f of scratch patterns was carried out with a microscope OLYMPUS GX51 (Košice, Slovakia) with a hundredfold magnifying effect. The results were processed by the program ImageJ which evaluated the percentage share of porosity in the examined location (The measurement was performed in Košice, Slovakia).

Assessment of the structures of the eutectic structures was carried out in the locations in the case where measuring of the selected mechanical properties was performed. It was the case of the critical locations of the castings, i.e., the structural holes in the body of casting. During die casting the cores are situated in these locations and on the basis of both assessment of the hydrodynamics principles and of liquid metal flowing around the cores they were assessed as the locations with high probability of occurrence of foundry defects in the casting. Figure 3 shows the locations of sampling, and the samples were used for assessment of metallographic structure. The quality of castings is influenced by the method of filling of shaping the mold cavity and the proceeding melt flow in the shaping mold cavity. Due to the aforementioned, the F samples were taken in the location occurring opposite to the gate in case of which the melt flow hits the mold flange and splits itself into two flows moving along the mold wall towards the gate. The R samples were taken in the locations in the case where the melt flow, reflected from the head of the shaping cavity, becomes blended with the melt flowing through the gate.

The locations of the scratch patterns were selected also with regards to diverse thermal procedures in the casting volume. The scratch patterns were made on the front part below the surface as well as in the cross section of the casting according to zones in the direction of the thermal gradient during cooling.

The scratch patterns of samples taken in the selected locations of casting were realized in accordance with the ČSN 42 0491 standard (in the Czech Republic the standard is given without the ISO equivalent). To highlight the structure the scratch patterns were etched with a 0.5% hydrous solution of hydrofluoric acid at a temperature of 22 °C. The images were evaluated by the microscope OLYMPUS GX51 (České Budějovice, Czech Republic) with a hundredfold magnifying effect. The scratch patterns were realized with the samples taken from the sets of castings labelled Sample 1, Sample 4, and Sample 5. Linear change of values of the examined properties could be observed in the case of samples ranging from Sample 1 to Sample 4. The samples with minimal and maximal values of the examined properties were selected. Sample 5 was selected because of local extremes occurring within the linearity of the examined parameters (The measurement was performed in České Budějovice, Czech Republic).

## 3. Results

The results obtained through the performed experiments can be divided into three parts as follows: evaluation of mechanical properties, evaluation of porosity, and evaluation of metallographic structure.

### 3.1. Analysis of Mechanical Properties

#### 3.1.1. Analysis of Permanent Deformation

Measuring of permanent deformation was performed in the case of 15 test samples. The achieved results are shown in Table 4.

#### 3.1.2. Analysis of Surface Hardness

The hardness values (HB) were measured in case of five locations of the selected and assessed samples in relation to gate height change. The results are shown in Table 5.

A remarkable difference in values of the surface hardness of the casting in the case of gate height change was not proved and thus it can be assumed that the influence of gate height change on values of hardness can be neglected.

### 3.2. Analysis of Porosity of Castings

Inner homogeneity of castings was assessed in the case of samples which served as test samples during analysis of permanent deformation. The measuring location in the case of the assessment of homogeneity was also the location of measuring the permanent deformation. It was a suitable alternative in order to detect the mutual correlation between the inner homogeneity of the castings and of permanent deformation Evaluation of porosity f was performed with the samples, the permanent deformation values of which to a high degree approximated the arithmetic average of deformation intended for the given set of samples according to Table 4. The porosity values measured in relation to gate height are given in Table 6.

Table 6 presents porosity values in the case of scratch patterns in the cross section made through the structural hole of the casting (Figure 1). Figure 4 shows the evaluation of the percentage content of pores in the area of the scratch pattern using the ImageJ program. Figure 4a shows the content of pores in the case of the 4B sample, the porosity values of which were the lowest ones. Figure 4b shows pores in the case of the 5A sample which reached the highest porosity content in the area of the scratch pattern.

### 3.3. Analysis of Metalographic Structure

The detection of the eutectic components percentage was realized using the program ImageJ. Table 7 represents the measured values of the α-phase fraction in the grinding of individual samples.

In Figure 5, the structure on the metallographic grindings plane on the samples taken from the location against the gate is presented.

Figure 6 presents a comparison of metallographic grindings and structure of the samples taken from the location near the outlet of the gate into the cavity mold.

Based on the comparison of the sample structures taken from the castings made with different gate heights, it can be stated, that at all of the observed locations a noticeable increase of the α-phase in the grinding plane with decreasing height of the gate is apparent. It is possible to claim that the change in the height of the gate has a direct effect on the structural composition of the eutectic.

## 4. Discussion

### 4.1. Evaluation of Mechanical Properties

#### 4.1.1. Evaluation of Permanent Deformation

The shape of the selected casting did not allow the production of testing bars designed for the statistic pull test. Therefore the pressure test was performed (permanent deformation s) and its measuring was carried out at the critical location of the casting (flowing around the cores), installation hole according to Figure 1. The lowest values of permanent deformation were detected in the case of samples which were made of castings having been die cast at the gate height of 0.82 mm. On the basis of such an observation it can be assumed that the values of permanent deformation depend on the gate height due to the modulation of the flow and speed of the melt passing through the gate. Then this determines the mold cavity filling mode. The presumption is that at the gate height of 0.82 mm the melt flow reaches such speed that it determines the mold cavity filling mode in combination with turbulent and disperse flow. This presumption has been proved and the influence of geometry of the gate on technological parameters is examined in the publication [22].

#### 4.1.2. Assessment of the Surface Hardness

The considerable difference between the measured surface hardness values dependent on gate height change within the framework of the performed analysis was not proved which is also clear from the obtained values. The analysis confirms the fact that the hardness of the castings depends especially on the casting structure and size of grains. The size of the crystal grains is derived from the level of undercooling of the melt when coming into contact with the mold and the speed of cooling of the melt. The determining factor of the grain size is the thermal gradient between the melt and the mold [4].

### 4.2. Analysis of Porosity of Castings

Porosity f was evaluated in the case of the test samples taken from the critical location of the casting by cutting perpendicularly in relation to the axis of the installation hole (Figure 1). The influence of the gate height affects the porosity values especially by shaping and directing the flow of the vmelt being forced into the shaping mold cavity and by the change of the melt speed flowing through different areas of the gate, i.e., by the change of the mold cavity filling mode [23].

### 4.3. Reasons for Structural Changes

Searching for the reasons of structural change it is possible to rely on the experimental research carried out by Borisov and Batyšev. One can hypothesize that pressure has a similar effect in the process of crystallization and solidification of Al-Si alloys castings as their modification. According to Batyšev´s research, the effect of high pressure evinces a never increasing eutectic temperature and the point of eutectic crystallization is shifted to a higher silicon volume. The eutectic temperature is increased by about 6.3 °C every 100 MPa and the maximum solubility limit of silicon in aluminum is shifted by about 0.25 weight percent of silicon at the eutectic conversion. With the increasing pressure, the diameter of the primary α-phase is descending, meaning that the structure is finer and the influence of the holding pressure increases with the increasing of the casting wall thickness. In terms of the reduction of the structural parameters, only qualitative changes occur due to the pressure in the structure. Reducing the volume of the eutectic in the Al-Si alloy against the equilibrium state while increasing the silicon concentration in the eutectic and refining the structure becomes more visible the higher the pressure value. Due to the shift of the eutectic point in the equilibrium diagram of the Al-Si system, the proportion of primary α-phase with increasing pressure decreases [3,10,17,19].

It is not possible to accept the above mentioned claim in full, since the technological parameters of the casting process were maintained by casting of each series of castings at a constant level. As mentioned above, the limiting factor is pressure. Therefore it is possible to hypothesize that the height of the gate affects the transfer of the hydrostatic pressure and the length of the holding pressure phase, which subsequently manifests on the internal structure of the casting. To verify this assumption, a series of simulations were performed in MAGMA5 version 5.3.1.3 where the input parameters were identical to the real process ones. To assure detailed examination it was inevitable to set up a fine distribution of the net. The number of cells is 52,635,960 and the number of cavity cells is 1,640,521. The dimensions of the cell 0.5 mm × 0.25 mm × 0.5 mm were set up for the gate. Such a fine set up of the network allows detailed examination of the assessed parameters.

The gate solidification time was selected as an evaluation parameter. The location where the temperature change was evaluated is shown in Figure 7.

The length of the holding pressure phase is the time from complete filling of the cavity mold to the time at which the temperature in the gate drops to the solid temperature of the alloy. For EN AC 47 100 alloy, the solid temperature is set to T_S_ = 560 °C. Table 8 shows the solidification times of the gates for each series of castings.

Based on the simulations performed, the evaluation of solidification in the area of the gate shows, that decreasing the height of the gate decreases the time of solidification. A gate with lower height solidifies in a shorter interval of the time, thus the influence time of the holding pressure also shortens. The difference between the solidification times with the extreme values of the gate heights is Δt = 0.149 s. As discussed above, the proportion of primary α-phase decreases with increasing pressure. Based on observations of the metallographic samples grindings taken from the volume of the casting and based on the evaluation of simulations, it is relevant to claim, that the proportion of α-phase in the eutectics decreases depending on the time of the holding pressure influence.

### 4.4. Analysis of the Structure Influence on Mechanical Properties of Castings

A eutectic is a mixture of a solid solution of α-phase and β-phase crystals resulting from eutectic conversion. The silumin eutectic is characterized by a concentration of 11.7 to 12.5% of silicon. The α- phase is a solid aluminum solution with different volumes of other elements excreted as white formations. The β-phase is a solid aluminum solution of almost pure silicon (containing over 98% of Si) excreted as grey formations. Silicon increases the strength of a solid solution and its corrosion resistance. With higher volume, it is present as a pure silicon, which increases the hardness but reduces the deformation characteristics and toughness. This can be associated directly with the permanent deformation values [22].

Comparing the structures of metallographic grindings taken from the castings with different heights of gates, the increase of a β-phase proportion in the grinding plane with increasing gate height was demonstrated. Since the grindings were performed on samples taken at different locations of castings, we can state, that the increase of β-phase is in the whole volume.

Comparing the average values of the mechanical properties of the individual series of castings listed in Table 4, Table 5 and Table 6 with the results in Table 7, the following conclusions can be formulated:The ratio of both the α-phase and the β-phase in the eutectic is directly dependent on the time of the holding pressure

With increasing of the gate height, the effect of the holding pressure over a longer period of time is more effective, resulting in a better eutectic conversion and the increase of β-phase in the entire casting volume.

The increase of a β-phase adversely affects the values of a permanent deformation

Since the β-phase consists of a proportion of silicon above 98%, its higher proportion causes a reduction in toughness and hence resistance to deformation due to the permanent load. Table 4 presents the decrease of the linear deformation within the range from Sample 1 to Sample 4. The certain local extreme is Sample 5. This fact can be explained by the proportion of porosity in individual specimens where specimens within Sample 5 evinced high porosity values. The cause of the extreme porosity formation is the cavity mold filling mode. The height of the gate, which is the lowest in these castings, provides acceleration to the melt flow giving a dispersive character to the cavity mold filling mode. Dispersion filling allows enclosing of gas in the melt volume and thus genesis of increased porosity.

An increasing proportion of β-phase affects the casting surface hardness values

As demonstrated, the casting surface hardness values are not dependent on the height of gate but on the degree of subcooling of the melt upon contact with the face of the mold. A minimum hardness difference ΔHB = 1 is detectable between Sample 1 and Sample 2 casting series. When compared to the metallographic grindings of the samples, it is clear that Sample 4 and Sample 5 have a lower proportion of β-phase than Sample 1. Silicon increases the hardness of the alloy, so its higher volume in the casting can influence the casting surface hardness.

### 4.5. Evaluation of Mutual Correlation of the Observed and Evaluated Factors

On the basis of the aforementioned and pursuant to the described experiments and expressed mutual relations between the observed and evaluated factors influencing the die casting process and the experiments described in the professional literature [1,2,3,4,5,6,7,8,9,10,11,12,13,14,15,16,17,18,19,20,21,22,23], the mutual correlation between the gate structure, the mechanical properties of the casting, its structural composition, and the technological parameters can be expressed as in Figure 8.

## 5. Conclusions

The submitted paper presented the results of a study concerning the influence of gate geometry on the possibilities of affecting the mechanical properties and the eutectic structure of silumin based castings. The achieved results proved that the gate height represents one of the basic structural factors which influences the qualitative properties of die cast castings and determines the speed and mold cavity filling mode.

It was proved that the value of permanent deformation corresponds with the porosity which reduces the cross section of the casting. A significant factor, apart from the size of the cavities, is also the location in the casting volume. It can be assumed that evenly distributed cavities of smaller dimensions reduce the mechanical properties to a lower degree contrary to cavities with relatively large dimensions or clusters of cavities with the same volume share in the casting volume. When comparing the values of the casting porosity f and the permanent deformation s in relation to change of gate height it can be stated that with the increase of gate height the permanent deformation s increases as well. The exception to the aforementioned is represented only by the value of 0.75 mm. With regards to increasing gate height the casting porosity f tends to increase as well with deviation in the case of the value of 0.75 mm. The results point out the fact that the filling mode of the shaping mold cavity is a combination of turbulent and disperse flow.

Comparison of the surface hardness values HB illustrates that the change of gate height does not have any relevant influence on hardness since its values depend on the alloy structure.

The causes of the changes in the eutectic structure morphology were presented and clarified and the influence of the components of the eutectic on the mechanical properties of the castings was evaluated as well. It was proved that the gate height can influence the share of the α phase in relation to the β phase in the eutectic in the entire volume of the casting. The monitored properties were subjected to confrontation with the change of share of the β phase. With the change of the share of the β phase in the volume of casting, the mechanical properties of the casting changed as well. Therefore there exists a justified reason to state that the share of the β phase influences the change of the mechanical properties, particularly it influences the permanent deformation values and to a certain degree even the values of surface hardness of the castings are also affected.

The paper presented the influence of gate height on formation of the eutectic alloy structure and its influence of the properties of the castings. It was proved that the formation of the eutectic structure is influenced not only by the height of hydrostatic pressure but by the holding phase duration as well. The period of gate solidification is thus one of the determining parameters in the case of the melt crystallization with the possibility of partial influence of the mechanical and qualitative properties of the castings.

The achieved results unambiguously proved that the gate height represents a significant structural dimension of the mold gating system because there occurs right in the gate itself the change of the melt flow speed and modulation of the melt flow leading to shaping the mold cavity. As a consequence, the flow determines the filling mode which influences the casting homogeneity.

On the basis of the achieved results the measures for the gating system structure were adopted with regards to corrections carried out after mechanical testing of the castings. The gate height had to be changed from a value of 1.25 mm to a value of 0.82 mm in the case where the values of the selected mechanical properties achieved the most advantageous parameters. These proposals and modifications will be implemented in the production process of the particular types of casting in the foundry plant.

## Figures and Tables

**Figure 1 materials-13-03539-f001:**
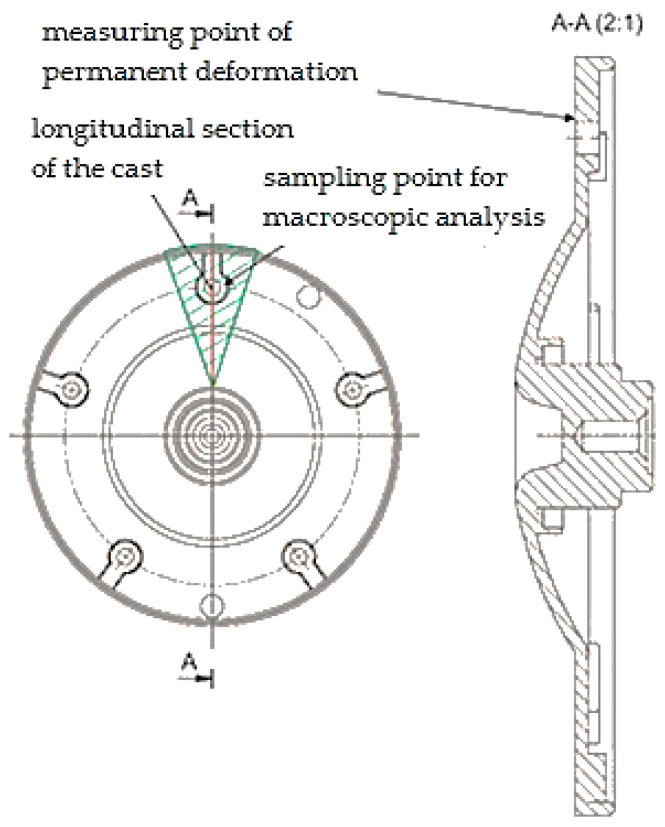
Measuring points of mechanical properties.

**Figure 2 materials-13-03539-f002:**
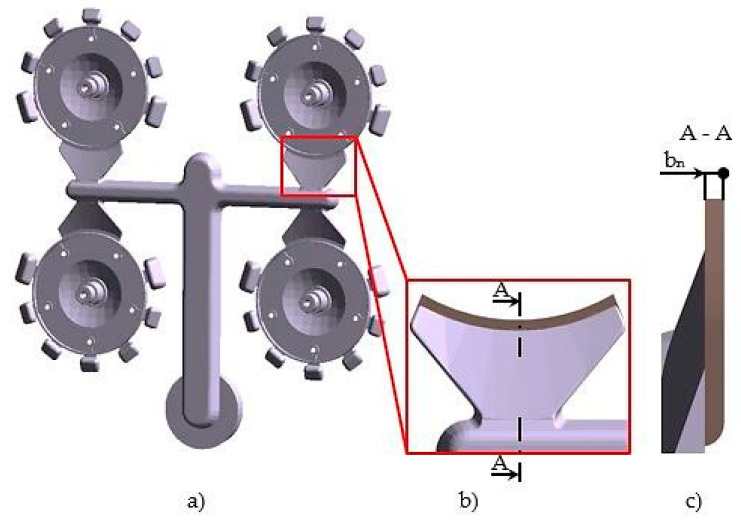
Structural design of the gating system and of the gate (**a**,**b**) the structural design of the connection of the casting to the gating system by means of a gate and its shape (**c**) the gate height.

**Figure 3 materials-13-03539-f003:**
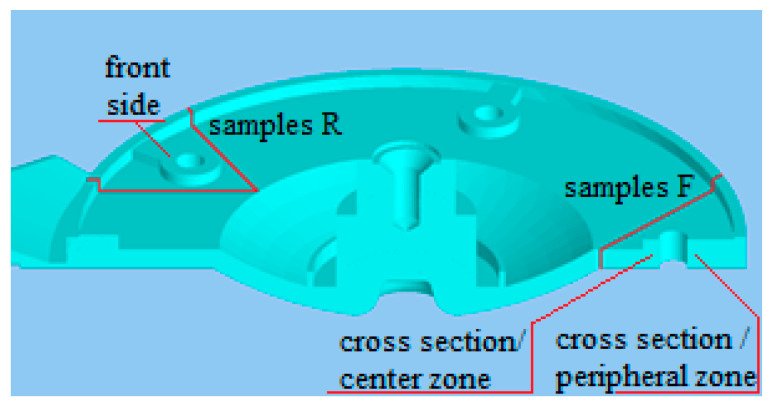
Evaluated locations of eutectic structures.

**Figure 4 materials-13-03539-f004:**
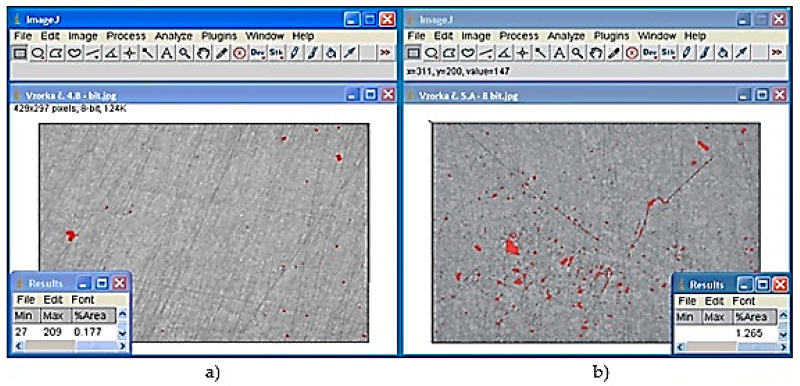
Evaluation of porosity by the ImageJ program (**a**) content of pores in the case of the 4B sample (**b**) content of pores in the case of the 5A sample.

**Figure 5 materials-13-03539-f005:**
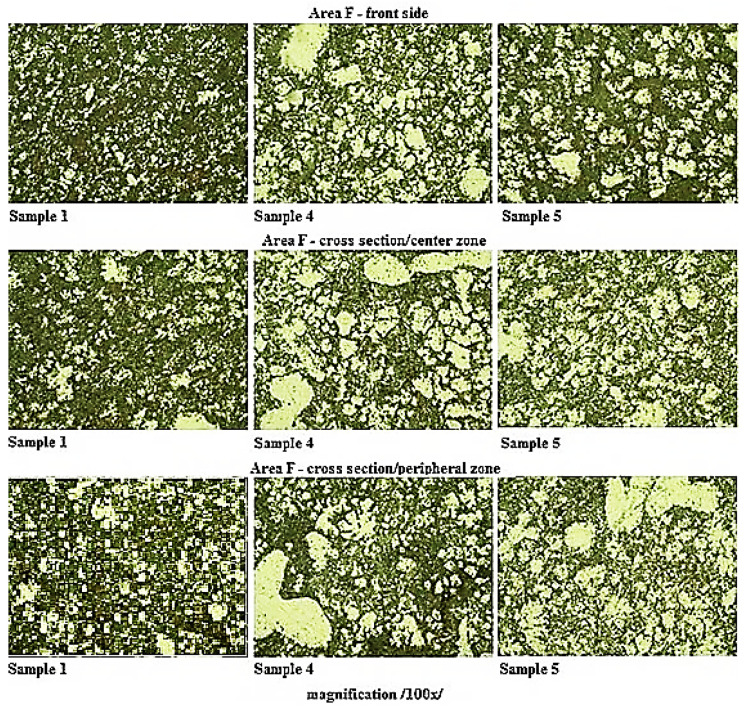
Structure in the location of the scratch pattern of samples taken from the F Area.

**Figure 6 materials-13-03539-f006:**
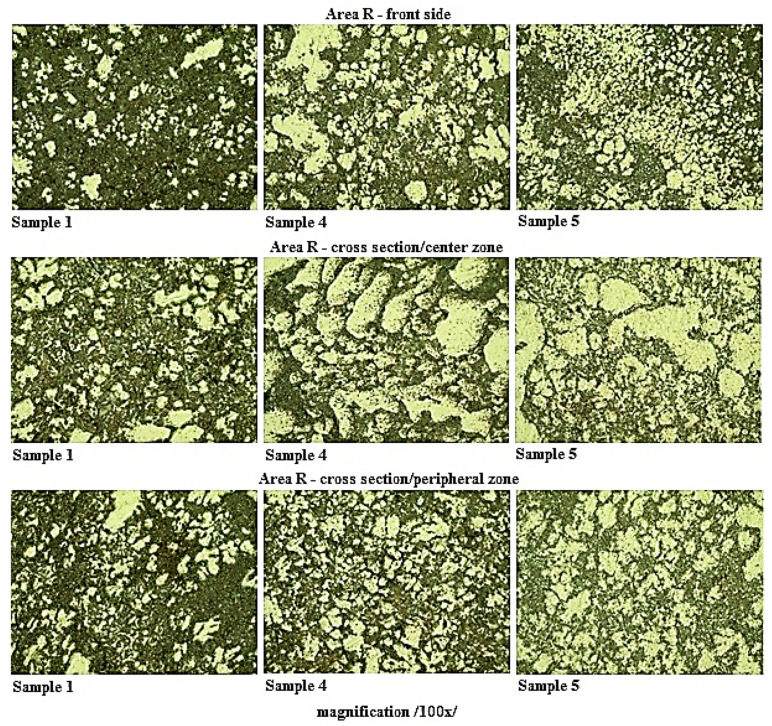
Structure in the scratch pattern location of samples taken from the R Area.

**Figure 7 materials-13-03539-f007:**
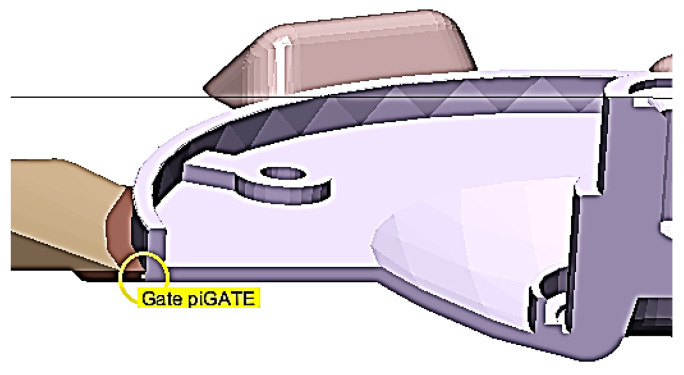
The location of the melt temperature evaluation in the gate.

**Figure 8 materials-13-03539-f008:**
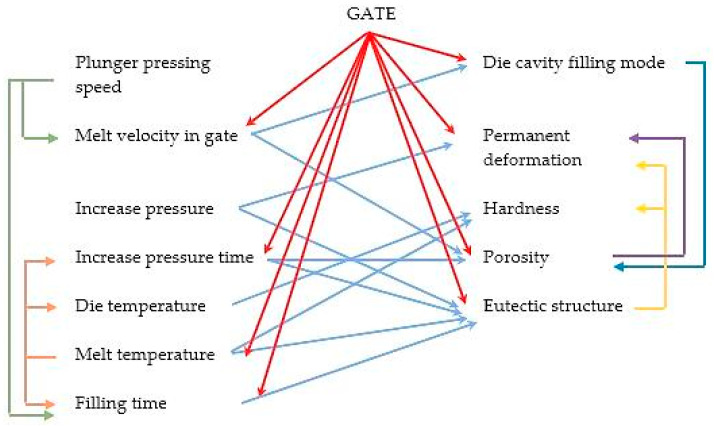
Mutual correlation of factors.

**Table 1 materials-13-03539-t001:** Chemical composition of alloy.

Chemical Composition of the Experimental Melt of the Used Alloy [%]											
Al	Si	Fe	Cu	Mn	Mg	Cr	Ni	Zn	Pb	Sn	Ti
85.27	12.02	0.71	1.19	0.21	0.13	0.02	0.02	0.35	0.02	0.03	0.03

**Table 2 materials-13-03539-t002:** Gate dimensions.

**Gate Height b, mm**	**b_1_**	**b_2_**	**b_3_**	**b_4_**	**b_5_**	**Gate Length a, mm**
	1.25	1.03	0.92	0.82	0.75	60.968
**Gate Area S_G_, mm^2^**	**S_G1_**	**S_G2_**	**S_G3_**	**S_G4_**	**S_G5_**	
	76.210	62.797	56.090	49.994	45.726	

**Table 3 materials-13-03539-t003:** Technological parameters of the casting cycle.

Technological Parameters of the Casting Cycle	
**Parameter**	**Value**
Temperature of the melt, °C	708
Temperature of the mold, °C	220
Speed of molding piston, m.s^−1^	2.9
Holding pressure, MPa	25
Filling time of the die cavity mold, s	0.019

**Table 4 materials-13-03539-t004:** Values of permanent deformation s.

Sample No.	Gate Height b, mm	Permanent Deformation s, mm		
			Average	Variance of Values
1.A	1.25	0.077	0.068	0.015
1.B		0.065		
1.C		0.062		
2.A	1.03	0.048	0.053	0.009
2.B		0.057		
2.C		0.055		
3.A	0.92	0.041	0.044	0.006
3.B		0.045		
3.C		0.047		
4.A	0.82	0.037	0.033	0.006
4.B		0.032		
4.C		0.031		
5.A	0.75	0.057	0.058	0.008
5.B		0.054		
5.C		0.062		

**Table 5 materials-13-03539-t005:** Values of surface hardness of casting HB.

Sample No.	Gate Height b, mm	Measuring					Average
		No. 1	No. 2	No. 3	No. 4	No. 5	
1	1.25	108 HB	107 HB	108 HB	107 HB	107 HB	**107 HB**
2	1.03	109 HB	107 HB	106 HB	107 HB	106 HB	**107 HB**
3	0.92	106 HB	108 HB	107 HB	108 HB	107 HB	**107 HB**
4	0.82	106HB	104 HB	107 HB	107 HB	107 HB	**106 HB**
5	0.75	105HB	107 HB	105 HB	106 HB	107 HB	**106 HB**

**Table 6 materials-13-03539-t006:** Porosity values f.

Sample No.	Gate Height b, mm	Porosity f, %
1.B	1.25	0.89
2.C	1.03	0.87
3.B	0.92	0.85
4.B	0.82	0.18
5.A	0.75	1.27

**Table 7 materials-13-03539-t007:** Percentage share of α-phase in the grinding.

	Percentage Share of α-Phase [%]		
**Area F**	**Sample 1**	**Sample 4**	**Sample 5**
front side	19.440	32.281	40.955
cross section/centre zone	24.931	41.390	46.186
cross section/peripheral zone	24.853	37.368	47.267
**Area R**	**Sample 1**	**Sample 4**	**Sample 5**
front side	19.540	39.093	42.376
cross section/centre zone	30.421	46.452	51.637
cross section/peripheral zone	26.614	37.621	43.799

**Table 8 materials-13-03539-t008:** Solidification time in the area of the gate.

Sample No.	Gate Height b, mm	Time of Solidification, s
1	1.25	0.354
2	1.03	0.278
3	0.92	0.264
4	0.82	0.207
5	0.75	0.205

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
