# Peer review of "Possibility of Utilization of Gate Geometry to Modify the Mechanical and Structural Properties of Castings on the Al-Si Basis"

_materials, 2020, doi:10.3390/ma13163539_

Round 1

Reviewer 1 Report

The aim of this manuscript is to study the effect of gate height on the mechanical properties and microstructure of Al-Si alloy and it might be contribute to the applications in engineering practice. Due to the following the consideration, I don't think the manuscript is acceptable:

  1. The experiments are not clearly descirbed on how to conduct the experiments on changes of the gates with different heights and necessary parameters or design consideration to evualte the gate speeds and filling time of the casting.
  2. The results on mechanical properties are not convincing, and the miscrostructures used for calculatte the percentage of the eutectics of the samples are not clear and not persussive.
  3. Simulation results and simulation conditions on how the consider the inference of the gate height to the microstructure of the casting are not enough.
  4. The conclusions are not clear.

Author Response

Dear Reviewer,

thank You for your comments.

Answers to your questions or comments:

Question or comment 1: The experiments are not clearly descirbed on how to conduct the experiments on changes of the gates with different heights and necessary parameters or design consideration to evualte the gate speeds and filling time of the casting.

Answer 1: It was modified. Line 136 - 205, line 301 - 310

Question or comment 2: The results on mechanical properties are not convincing, and the miscrostructures used for calculatte the percentage of the eutectics of the samples are not clear and not persussive.

Answer 2: It was modified. Line 167 - 179, line 192 - 205, line 301 - 310

Question or comment 3: Simulation results and simulation conditions on how the consider the inference of the gate height to the microstructure of the casting are not enough.

Answer 3: The aim of the simulation is to show what happens during the casting cycle. To understand the thermal processes and determine the solidification times, I think it is sufficient. The explanation of the change in the microstructure is in line 285 - 365.

Question or comment 4: The conclusions are not clear.

Answer 4: It was modified

Reviewer 2 Report

This paper investigated the correlation of Gate geometry, particularly the cast height for casting Al-Si basis alloy and its mechanical and structural properties. The deformation, surface hardness, porosity, and the metallographic structure of the samples were measured and correlated with the cast height. It is found out that the gate height determines the formation of eutectic alloy structure but not directly on the hardness. A lot of correlations are presented in this study which needs more in-depth understanding and discussion.

One question needs to be addressed: the gate area also changes with gate height, why the authors think the height is the controlling factor? How about the gate shape?

Author Response

Dear Reviewer,

thank You for your comments.

Answers to your questions:

Question 1: One question needs to be addressed: the gate area also changes with gate height, why the authors think the height is the controlling factor? How about the gate shape?

Answer 1: Structure of the gate stems from methodology described in publication . Influence of Structure Adjustment of Gating System of Casting Mould upon the Quality of Die Cast. The gate length for the respective type of casting is constant following the aforementioned publication and its methodology of structure of gate connection to casting with cylindrical area. From the point of view of structure the only parameter to influence the filling mode of mold shaping cavity is gate area SG. Since the gate area is in this case the function of width and height of the gate and the width is constant according to Influence of Structure Adjustment of Gating System of Casting Mould upon the Quality of Die Cast, the gate height bn was selected to be the examined parameter 

This answer is given in the article in lines 140 - 152

Reviewer 3 Report

Dear authors, I have critical questions and comments for you:

  1. There is some inconsistency in writing style. The document should be edited by an English editor for clarity.

  1. In general, the standards to be used should be ISO/ASTM, in case you cannot use these standards you should justify why.

  1. To facilitate reading and better understanding of the document, the Discussion and Results sections could be joined.

  1. The Materials and Methods section should contain all the equipment and procedures used. Paragraphs describing equipment outside this section should be reordered. For example, line 198, 215, 216 225 and 289.

  1. The influence of the location and height of the gate on the properties of the casting is discussed in this research. A figure showing the complete filling system should be incorporated. Also, it is of vital importance to know the geometry of the gate and its cross section. Table 2 details the parameter b that should be reflected in some figure. In addition, the parameter b has no reference and the values of the heights are not clear.

  1. Line 160: Why is it related porosity to deformation?

  1. What was the procedure for analyzing porosity? Is being evaluated surface porosity and pore shape? It is advisable to present examples of the pictures treated with the ImageJ.

To analyze the volumetric porosity, Micro-CT or Archimedes method techniques are needed. Table 6 shows results of surface porosity? If so, is it the percentage of area occupied by the pores? It is recommended to report the total area examined to analyze porosity. This section needs further analysis and explanation.

  1. Figure 3 is of very low quality.

  1. Line 270: every time a paper is named must be referenced.

  1. Line 289: it is necessary to justify the simulation with several points.

- Number of nodes in the mesh.

- Convergence method used for the resolution of the nodes.

- Thermal transmission coefficient parameter for the mould and the alloy used.

It should be reinforced with images of the simulation that justify the data obtained in Table 8.

  1. Line 309: this paragraph does not provide information for the investigation.

  1. The conclusions are inconsistent, the response of the input height is not analyzed and the best configuration based on the results obtained is not explained.

Many elements of the work are valuable. The article needs to be improved.

Author Response

Dear Reviewer,

thank You for your comments.

Answers to your questions or comments:

Question or comment 1: There is some inconsistency in writing style. The document should be edited by an English editor for clarity.

Answer 1: The document was edited by an English editor.

Question or comment 2: In general, the standards to be used should be ISO/ASTM, in case you cannot use these standards you should justify why.

Answer 2: The standards have been adapted to ISO Or the reason for the compensation explained. Line 167 - 174, line 198

Question or comment 3: To facilitate reading and better understanding of the document, the Discussion and Results sections could be joined.

Answer 3: In accordance with the template, the parts were left divided

Question or comment 4: The Materials and Methods section should contain all the equipment and procedures used. Paragraphs describing equipment outside this section should be reordered. For example, line 198, 215, 216 225 and 289.

Answer 4: It has been modified. Line 171 - 179

Question or comment 5: The influence of the location and height of the gate on the properties of the casting is discussed in this research. A figure showing the complete filling system should be incorporated. Also, it is of vital importance to know the geometry of the gate and its cross section. Table 2 details the parameter b that should be reflected in some figure. In addition, the parameter b has no reference and the values of the heights are not clear.

Answer 5: A figure showing the complete filling system is added to the article - Figure 2. On the Figure 2 is shown the profile of the gate - b) c) and its parameters.

Question or comment 6: Line 160: Why is it related porosity to deformation?

Answer 6: This fact is explained in line 225 - 232

Question or comment 7: What was the procedure for analyzing porosity? Is being evaluated surface porosity and pore shape? It is advisable to present examples of the pictures treated with the ImageJ.

Answer 7: This fact is explained in line 234 - 240.  In articel is added Figure 4 Evaluation of porosity by the ImageJ programme.

Question or comment 8: Figure 3 is of very low quality.

Answer 8:  It has been modified. Now it is Figure 5

Question or comment 9: Line 270: every time a paper is named must be referenced.

Answer 9: It has been modified. Line 271

Question or comment 10: Line 289: it is necessary to justify the simulation with several points

Answer 10: It has been modified. Line 301 - 310

Question or comment 11: Line 309: this paragraph does not provide information for the investigation.

Line 309 - Now Line 321: this section provides information on the time of holding pressure application

Question or comment 12: The conclusions are inconsistent, the response of the input height is not analyzed and the best configuration based on the results obtained is not explained.

Answer 12: It has been modified. Its added section line 412 - 416

Round 2

Reviewer 3 Report

The research is well structured and the conclusions are supported by the results. There are no additional comments.